# A Fresh Look at Mouthwashes—What Is Inside and What Is It For?

**DOI:** 10.3390/ijerph19073926

**Published:** 2022-03-25

**Authors:** Dominik Radzki, Marta Wilhelm-Węglarz, Katarzyna Pruska, Aida Kusiak, Iwona Ordyniec-Kwaśnica

**Affiliations:** 1Department of Periodontology and Oral Mucosa Diseases, Faculty of Medicine, Medical University of Gdańsk, 80-208 Gdańsk, Poland; aida.kusiak@gumed.edu.pl; 2Division of Molecular Bacteriology, Institute of Medical Biotechnology and Experimental Oncology, Intercollegiate Faculty of Biotechnology, Medical University of Gdańsk, 80-211 Gdańsk, Poland; katarzyna.pruska@gumed.edu.pl; 3Department of Dental Prosthetics, Faculty of Medicine, Medical University of Gdańsk, 80-208 Gdańsk, Poland; marta.wilhelm-weglarz@gumed.edu.pl (M.W.-W.); iwona.ordyniec-kwasnica@gumed.edu.pl (I.O.-K.)

**Keywords:** mouthwashes, oral hygiene, oral diseases, sodium bicarbonate, stannous fluoride, essential oils, glycerine, ethanol, zinc compounds

## Abstract

Mouthwashes are a very popular additional oral hygiene element and there are plenty of individual products, whose compositions are in a state of flux. The aim of our study was to investigate the compositions of mouthwashes and their functions, as well as to discuss their effectiveness in preventing and curing oral diseases and side effects. We searched for mouthwashes available on the market in Poland. We identified 241 individual mouthwash products. The extraction of compositions was performed and functions of the ingredients were assessed. Then, analysis was performed. The evaluation revealed that there are plenty of ingredients, but a typical mouthwash is a water–glycerine mixture and consists of additional sweetener, surfactant, preservative, and some colourant and flavouring agent, as well as usually having two oral health substances, anticaries sodium fluoride and antimicrobial essential oils. The effectiveness or side effects of several substances of mouthwashes were thoroughly discussed. We recommend not multiplying individual mouthwash products and their ingredients beyond medical or pharmaceutical necessity, especially without scientific proof.

## 1. Introduction

Since the disseminated change from hunter-gatherer to agrarian societies and agriculture development, dental caries have been observed to be more frequent. These are associated with the alteration in human diet and food preparation techniques: meals became more farinaceous, rich in simple sugars, soft and adherable. The more refined plant foods do not stimulate saliva flow so efficiently and do not ensure self-cleaning as their unrefined counterparts, which worsen buffering capacity and the remineralisation of enamel and support dental plaque accumulation [1,2,3,4]. With the development of modern healthcare and access to various foods, periodontal diseases related to vitamin C deficiency are almost forgotten, but the ineffective removal of dental plaque in association with chronic stress, inveterate smoking, diabetes, dyslipidemia, etc., may result in periodontitis, which affects a large part of the population worldwide [2,5,6,7,8]. Rising life expectancy generates other problems such as tissue degeneration as well as neoplasms and their treatment, which may result in hyposalivation, immunodeficiency or malnutrition with subsequent oral diseases [9,10,11,12]. Proper oral hygiene is the basis of keeping the oral cavity healthy. Brushing and flossing, as well as a suitable diet and hydration, are all necessary. Nevertheless, there are additional, adjuvant oral hygiene products to potentiate the effectiveness of oral hygiene, such as irrigators, tongue scrapers and mouthwashes.

Mouthwashes are a very popular additional oral hygiene element and there are plenty of individual products. There are two main types of mouthwash applications: preventive and therapeutic. A single product may possess a double function: antiplaque substances prevent as well as support the treatment of periodontal diseases, among others. It is not strictly associated with the concentration of active substances seen in mouthwashes, but with the duration of usage and of course with current health status. In prevention, long-term use is required, whereas in therapy, short-time use is usually sufficient. Another function is to provide relief in some conditions, in preoperative or postoperative management as well as in aesthetic dentistry (anti-stain and whitening effects). 

Mouthwashes have been under study for a long time and their compositions are in a state of flux. A substance once considered to be promising may today be forgotten, while others are still the standard of prevention and therapy [13]. The aim of our study was to investigate the compositions of mouthwashes and their functions.

## 2. Materials and Methods

### 2.1. Definition of Mouthwashes 

Mouthwashes (also called mouth rinses/mouthrinses, oral rinses or oral washes) are liquid, aqueous compositions mainly intended to prevent, relieve and cure oral conditions and maintain oral health (such as: dental caries, dental erosion, halitosis, gingivitis, periodontitis, mucositis, to reduce the oral microbiota, etc.). Our study is restricted to over-the-counter compositions. Saliva substitutes are not taken into account, as they are a group of special-purpose drugs, other than mouthwashes, used when own production of saliva is inadequate. However, indications of those two groups may overlap.

The mouthwashes were selected according to the following inclusion criteria: Intended to prevent and cure oral conditions and maintain oral health;Over-the-counter;In one of the following forms:
An aqueous solution;A concentrate to be diluted with water;A tablet or powder to be dissolved in water.

The exclusion criteria were as follows: Full composition not available;Saliva substitutes;Not recognised as cosmetic;Containing antibiotics, antimycotics, steroids, parasympathomimetic saliva stimulants, topical local anaesthetics and other prescription drugs;Orally taken, but not intended for oral health (e.g.,nitroglycerine)Toothpastes, gels, powders, foams and similar;With majority of coconut oil or other oils;Homeopathics;Tablets for chewing;Intended to cure tonsillitis, pharyngitis and other conditions of the respiratory tract.

### 2.2. Search Strategy and Data Collection

We searched for mouthwashes available on the (online and traditional) market in Poland, including online stores, which may correspond to markets in other countries (trade names or ingredients may differ). The authors used browsing methods available to an average consumer. Search, identification and extraction of compositions were performed by two researchers (D.R. and M.W.W.) independently. Only readily available (package/website) data on the product’s composition were used in the study; we did not contact the producer to request missing data (e.g., concentration of fluorine). Subsequently, the lists of mouthwashes and their compositions were compared. In case of disagreement, the composition was checked and corrected. Then, the two lists were combined. The third researcher (K.P.) checked the fused list. As compositions may be changed by producers without notice, there may exist differences between declared and actual composition of a product. In our study, we worked with the assumption that the data provided by the producers is accurate. We only analysed products available on the market as of the time of conducting this study (November 2020–February 2022), and some of them may have since been discontinued.

### 2.3. Analysis

After identification and extraction of compositions, functions of all mouthwash ingredients were established and listed. Then, several groups of ingredients were distinguished according to their main functions. Among the groups, an analysis of frequency of substances was performed. 

## 3. Results

### 3.1. Components and Their Functions

We focused on the chemistry and medical significance of the ingredients used in mouthwashes, with the exception of the majority of plant extracts (due to their number and very limited clinical research on herbal-based dentifrices [14]). Ingredients and their properties (in mouthwashes) were tabulated and presented in Table 1 (Ingredients of mouthwashes and their main functions therein). There is no certainty that an indicated property is observed in the mouthwashes, as for each composition there are many factors at play that may alter the characteristics of the substance. It appears that some of the components may present no effect (neither medical nor pharmaceutical) and are used only for marketing purposes. Their properties were mostly described in isolation; potential interactions were not explored. Potential of hydrogen, viscosity, electrical conductivity, the volume of mouthwash, concentration of each ingredient, mutually exclusive properties or synergies of ingredients, time of rinsing, chemical reactions between ingredients (e.g., between anions and cations or chelation), time of storage, etc., may result in some changes in properties. 

### 3.2. Analysis of Mouthwash Components

We identified 241 individual mouthwash products according to the inclusion and exclusion criteria, of which 234 were aqueous solutions and 7 in the form of a tablet. The full composition of each mouthwash is presented in the supplementary material (Appendix A). Among them, several groups were distinguished according to their main postulated functions (functions of substances from each group may overlap to some extent). We subdivided them as follows: oral health substances, solvents, surfactants and thickeners, sweeteners, plant (extracts, waters and oils), preservatives, colourants, flavouring or cooling agents, and others.

Among oral health substances, several groups were recognised. Fluorine compounds are present in 63.5% of all mouthwashes (individual products), and the most common compound is sodium fluoride (86.9% of mouthwashes containing fluorine compounds) with a concentration around 187–250 ppm of fluoride ions. Arginine as well as sodium bicarbonate are present in 1.2%, and aluminium lactate in 1.7%. Potassium compounds are present in 17.8% of them, and tetrapotassium pyrophosphate is the most common potassium compound, followed by potassium chloride and potassium nitrate. Zinc compounds are observed in 19.5% of all mouthwashes; zinc chloride and zinc citrate are the most common. Phosphorus and calcium compounds are present in 17.8%, among them tetrapotassium pyrophosphate, tetrasodium pyrophosphate and hydroxyapatite are the most common. Stannous compounds (chloride and fluoride) are in 1.7% of mouthwashes. Antimicrobial drugs are very common and present in 52.3% of mouthwashes, therein usually essential oils, cetylpyridinium chloride or chlorhexidine. Essential oils are present in 47.3% of mouthwashes and the most common compound derived from essential oils is menthol, seen in 78.9% of mouthwashes with essential oils. Chlorhexidine is in 20.7% of mouthwashes and the usual compound is chlorhexidine digluconate (98% of mouthwashes with chlorhexidine) at a concentration of 0.2% (26%).

Mouthwashes are composed of many other substances, not directly associated with medical benefits, but necessary in the creation of formulas. Ethanol, glycerine and propylene glycol are the most frequent solvents, except water, present in 10.8, 74.7 and 42.7% of all mouthwashes, respectively. Surfactants are in 92.5% of mouthwashes, and PEG-40 hydrogenated castor oil is the most frequent. Sweeteners (together with glycerine) are present in 96.7%, with saccharin, sorbitol and xylitol being the most common (apart from glycerine). Preservatives are found in 82.2%, with the majority being sodium benzoate. Colourants are added to 66% mouthwashes, while flavouring agents to 99.2%. Entire results and details are presented in Table 2.

A typical mouthwash on the market is a water–glycerine mixture, consisting of an additional sweetener (saccharin), surfactant (PEG-40 hydrogenated castor oil), preservative (sodium benzoate), some colourant and flavouring agent, as well as having two oral health substances, an anticaries compound (sodium fluoride, around 217–250 parts per million) and antimicrobial drug (essential oils).

## 4. Discussion

Selected ingredients found in mouthwashes and their uses in hygiene of the oral cavity are discussed below. 

### 4.1. Sodium Bicarbonate

Sodium bicarbonate is also known as baking soda, bicarbonate of soda, and by the IUPAC name, sodium hydrogen carbonate. Sodium bicarbonate is a versatile substance; there are many general and medical uses of it. It is used in the treatment of metabolic acidosis and acid reflux, and as a mucolytic agent [84]. Sodium bicarbonate is a nonspecific antidote that is effective in the treatment of a variety of poisonings [85].

It is composed of a sodium cation (Na^+^) and a bicarbonate anion (HCO_3_^−^), which are normal constituents in the human body. The bicarbonate anion is the result of interconversion of water and carbon dioxide. Carbonic anhydrases (CAs) are a family of several zinc-dependent enzymes which increase the efficiency of bicarbonate anion formation. CAs maintain the excretion of carbon dioxide from the body as well as acid–base homeostasis [86]. Only one is isoenzyme of CAs, CA VI (called gustin), produces a secreted form of the bicarbonate anion. CA VI is produced by serous acinar cells of parotid and submandibular glands [87]. It is also detected in tears and milk [88]. Its primary function is to regulate pH in the oral cavity. Bicarbonate is the principal buffer of extracellular fluid (plasma and interstitial fluid) as well as saliva. When fermentation of carbohydrates in the oral cavity occurs, pH lowers (increasing concentration of H^+^), but is neutralised by the bicarbonate anion (by conversion to water and carbon dioxide) [84]. The supplementing of bicarbonate ions, introduced by rinsing with sodium bicarbonate solution, may be a sufficient refill of buffer capacity of saliva, when used regularly.

CA VI expression (both its concentration and activity) and bicarbonate anion concentration are supposed to be modified by various factors: genetic (single nucleotide polymorphism in the gene coding CA VI), chemical and a changed flow rate of saliva, etc. [89,90]. A mean CA VI concentration is significantly lower in the active caries group than in the caries-free group [89]. When the activity of CA VI is measured in the caries group, it is increased, but there is no increased ability of CA VI to buffer and buffering capacity of saliva is lower in the caries group [91]. Moreover, the ability of CA VI to buffer is more related to its concentration than its activity, but elevated CA VI concentrations in saliva do not necessarily indicate that the enzyme is active [92]. CA VI is a potential drug target for cariogenesis. Further studies are required.

A 1.26% (*w*/*v*) solution of sodium bicarbonate concentration is isotonic. Slightly hypotonic, 1% solution, can be used as a mild mouthwash to neutralise acids, moisturise, deodorise and decrease the number of bacteria [84]. A hypertonic solution of sodium bicarbonate facilitates osmotic movement of water from bacterial cells, resulting in shrinkage, plasmolysis and cell death [93]. Sodium bicarbonate in solution is able to disrupt biofilms without an antimicrobial effect, hypothetically by disrupting the exopolysaccharide matrix structure of dental plaque [94]. Hypernatremia after regular rinsing is unexpected if the product is not swallowed.

According to our study, some mouthwashes in tablets contain sodium bicarbonate. Nevertheless, therein sodium bicarbonate is just a component of effervescent salts (with citric acid or/and tartaric acid) [33]. After dissolution in water, neutralisation occurs—sodium bicarbonate through interaction with organic acids generates carbon dioxide and bicarbonate anions no longer exist. In addition, in our unpublished study, the pH of one of the tested products (Georganics Spearmint) was slightly acidic (pH 6.5; tablets dissolved in tap water pH 7.5). Therefore, we cannot consider those mouthwashes as a source of bicarbonate anions able to neutralise acidic pH of the oral cavity.

### 4.2. Essential Oils

Essential oils (EOs) are liquid, volatile, fragrant, limpid and coloured, soluble in lipids and organic solvents. EOs are derived from many plants and play an important role in protecting the plant, repelling predators and pests, as well as attracting pollinators. EOs consist of various compounds, predominantly monoterpenes (or monoterpenoids), sesquiterpenes, aromatic compounds (often phenylpropane derivatives, e.g., phenylpropanoids) and their derivatives: acids, alcohols, ketones and aldehydes, aliphatic hydrocarbons, acyclic esters or lactones, nitrogen- and sulphur-containing compounds, coumarins [95,96].

Essential oils contain a wide variety of secondary metabolites that exhibit antibacterial (bactericidal or bacteriostatic), antiviral and antifungal properties [95]. EOs or their components are regularly used in many dentifrices and mouthwashes due to their germicidal effects, resulting in a noticeable reduction of dental plaque, with anticaries [97] and antigingivitis potential [98]. They may also serve as preservatives. Antioxidant, anti-inflammatory, local anaesthetic and analgesic effects are other relevant properties [99,100]. Nevertheless, they play an important role in oral hygiene as fragrances.

EOs, as was said above, are composed of many substances, but only few of those show significant antimicrobial (antibacterial, antifungal and antiviral) [101] properties. The antimicrobial effects of EO components are not attributed to a unique mechanism, but rely on toxic effects on membrane structure and the entire cell. Due to their lipophilic character, they pass from the aqueous phase into membrane structures, which results in the disturbance of the cell membrane. Alteration of the membrane fatty acids, increased membrane fluidity and permeability, disturbance of membrane-embedded proteins, inhibition of respiration, alteration of ion transport processes and alteration of proton motive force are all observed. It is suggested that the cytoplasm is a secondary target. Compounds of EOs after penetration into cell plasma may interact with intracellular structures, which leads to cytoplasm coagulation, denaturation of several proteins and the loss of metabolites and ions, resulting in alterations in functionality. Most EOs have a more powerful effect on Gram-positive than Gram-negative bacteria, presumably due to differences in the cell membrane composition (presence of outer membrane in Gram-negative) and permeability to lipophilic substances [95,102,103].

Menthol, thymol, eucalyptol, eugenol and methyl salicylate (or EOs containing them) are regularly used in dentifrices and mouthwashes. There are several (clinical and microbiological) studies that demonstrated antibacterial effects of the mouthwashes containing EOs (ethanol and propane-1,2-diol based solutions). In spite of this, there is no evidence methyl salicylate in isolation possesses any antibacterial effect, but it is used as a fragrance, with anti-inflammatory effect [104]. Bisabolol possesses anti-inflammatory [105] and antimicrobial [106] activity. Menthol presents moderate antimicrobial (antibacterial) effects (*Mentha piperita* oil also shows antiviral and antifungal properties) [107]. Eucalyptol (1,8-cineole) [108,109], eugenol [110,111], and thymol [112] are effective antibacterial substances and present antibiofilm (inhibitory or disruptive) activity. Those substances also act as preservatives, to prevent foodborne pathogens in storage [113]. Antimicrobial components of EOs are usually used in complex compositions, with an expected wider antibacterial spectrum.

Moreover, there are plenty of other substances with antimicrobial activity derived from essential oils. Many of them are components of mouthwashes, both single compounds and essential oils. They are mainly used as a scent and there is no sufficient evidence that they act as antibacterial drugs intended for combating the oral microbiota (or even as preservatives), but there are such possibilities. Benzyl alcohol is mainly used as a fragrance and preservative [114,115] and only its derivative (2,4-dichlorobenzyl alcohol) is used as a mild oral and pharyngeal antiseptic. Limonene [99,116,117], linalool [99,117], citral [118] and cinnamal (cinnamaldehyde) [119] exhibit a non-negligible activity against oral pathogenic microorganisms. Geraniol [120] and anethole [121] also show some antimicrobial properties.

In our study, according to the established customs in (dental) studies and uses in cosmetics, we tabulated (in Appendix A and Table 2) EOs or their components in the following groups: 1. oral health substances (menthol, thymol, eucalyptol, methyl salicylate, additionally eugenol and bisabolol or EOs that are expected to contain a significant amount of them); 2. plant extracts, waters and oils (with potential various functions in mouthwashes; they are not under our study); 3. Preservatives (benzyl alcohol); or 4. aroma or cooling agents (every EO or its components tabulated in groups 1–3 may also act as a fragrance as well as every EO tabulated as an aroma may possess relevant antimicrobial properties).

### 4.3. Fluorine Compounds

Fluorine, specifically some compounds of fluorine, have become a principal agent in preventing dental caries. In mouthwashes, the most common compound of fluorine is sodium fluoride, others are calcium fluoride (uncertain solubility in mouthwashes), potassium fluoride, stannous fluoride (unstable in mouthwashes), monofluorophosphate, aluminium fluoride, fluorohydroxyapatite (as suspended nanoparticles), olaflur and nicomethanolhydrofluoride. The mechanism of action depends on their solubility and ability to release free fluoride anions. Cations may be responsible only for solubility (sodium fluoride) or confer additional properties (olaflur, stannous fluoride, aluminium fluoride). Sodium fluoride is a thoroughly researched compound: comprehensive and critical descriptions are commonly available, and its usage and benefits are well established, therefore we did not review them. Other fluorine compounds are relatively uncommon in mouthwashes. Nevertheless, we would like to describe one—stannous fluoride, a compound of fluorine that is potentially very useful in mouthwashes, but also very unstable.

Stannous fluoride, IUPAC tin (II) fluoride, shows a similar or slightly higher anticaries activity compared with non-stannous fluoride. Stannous fluoride mechanism of action in the prevention of caries is similar to sodium fluoride. Fluoride ions are deposed on the tooth surface as a layer of calcium fluoride. When the pH of saliva decreases, fluoride ions are released and can be incorporated into hydroxyapatite by substituting hydroxyl groups in hydroxyapatite, allowing the formation of fluorohydroxyapatite or fluorapatite, much more resistant to the acidic pH [122,123,124].

Stannous fluoride can be useful not only in the prevention of dental caries. Antibacterial effects of stannous fluoride were observed as well as its ability to prevent dental plaque formation resulting in antigingivitis property [125,126]. Stannous fluoride reduces calculus formation (by inhibition of calcium phosphate compound creation), halitosis (by reduction in volatile sulphur compound levels and antimicrobial activity) and dentinal hypersensitivity (by forming precipitates in the dentin tubules) [122,127,128,129,130]. Antimicrobial and anti-halitosis effects of stannous fluoride may be potentiated by content of zinc compounds, which are added to dentifrices as stabilisers protecting stannous from oxidation [129]. Stannous fluoride (as well as stannous chloride) can be useful in the prevention of enamel erosion [38,123]. When the demineralisation of enamel and dentin by extrinsic acids takes place, the organic matrix containing collagen is exposed. The exposed collagen can be degraded by endogenous (from dentin or saliva) collagenases, including matrix metalloproteinases (MMPs), which blocks its remineralisation. Stannous ions cause a direct inhibition of the MMPs, and the presence of both calcium and phosphate, with or without fluoride ions, promotes the remineralisation [67].

Those properties cannot be observed in commercial mouthwashes. The aqueous solution of stannous fluoride is unstable. Its ions are readily removed from solution by oxidation to stannic ions, by the formation of complex ions and by hydrolysis, thus becoming ineffective in desired therapeutic benefits [131]. (Stannous can be hydrolysed into SnOH^+^, Sn(OH)_2_^0^ and Sn(OH)_3_^−^at low concentration, Sn_2_(OH)_2_^2+^ and Sn(OH)_4_^2+^ are predominant at higher concentration [132]).The instability of stannous fluoride in water depends on time; therefore, only a freshly made solution should be used [131,133]. When trying to obtain a stable solution of stannous fluoride, the majority or, ideally, all of the water should be replaced by another solvent. Recently, a water-free concentrated solution of stannous fluoride has been released; it has to be mixed with water immediately before use and should not be stored in dissolved form, according to the manual. Stannous fluoride (and stannous chloride) may be solubilised in such low levels of water compositions containing low-polar solvents (such as glycerine and propylene glycol), forming non-ionic molecular structures. Such compositions remain stable during storage, with fluoride and stannous ions being released from the non-ionic structure when in contact with water or saliva [134,135]. This may not be the case for mouthwashes in the form of aqueous solutions. It is, however, possible for this compound to be present in a viable product—in the form of tablets (or a water-free concentrate) intended for preparing a fresh aqueous solution before every use. There are also formulas of water-free glycerine-based gels [136].

### 4.4. Glycerine

Glycerol (commonly called glycerine or glycerin, IUPAC propane-1,2,3-triol) is one of the most prevalent ingredients of the mouthwashes, but to our knowledge there are not many scientific papers focusing on its biological effects. Glycerol acts as a solvent and increases the solubility of some components in water, therefore it can be a substitute (to some extent) of ethyl alcohol and can also be used in formulation with ethyl alcohol and water together [137,138]. It is commonly used as a lubricant and emollient, as it provides smoothness, but due to its hygroscopic properties which may cause desiccation [139].

Due to its sweet taste, it may serve as a sweetener and improve the taste of mouthwash by masking the unpleasant taste of some ingredients. Its taste may promote an increased flow of saliva, as other sweeteners do [140].

Dental erosion is one of the most reported side effects of mouthwashes. The erosive potential of some mouthwash ingredients not only depends on their pH, but also on their viscosity. Glycerine elevates the viscosity of solutions, resulting in the ability of the liquid to adhere to the tooth surface and potentiate erosive properties [141]. Other factors, such as salivary flow and buffering capacity, might influence the overall effect of mouthwashes on the enamel [142]. Glycerol used in the formulation of isopropanol-based surgical hand rubs decreases the bactericidal efficacy [143]—the impact of glycerol on antibacterial properties of mouthwashes should be investigated. An in vitro study revealed that glycerol affects the mucosa homeostasis (increasing epithelial thickness, proliferation and apoptosis) if glycerol concentration is above 42.5% and does not affect mucosal integrity. Affected homeostasis and increased proliferation may result in DNA damage accumulation and pose potential risks in long-term exposure [139]. Glycerine is also an ingredient in the liquid used in electronic cigarettes, with no toxicologically relevant effects [144].

### 4.5. Ethyl Alcohol

Ethyl alcohol (EtOH), IUPAC ethanol, commonly and inappropriately just alcohol, is used as a polar solvent in mouthwashes, as well as a solubiliser, stabiliser, flavouring agent and preservative. It also displays an antiseptic effect. EtOH enhances the effect of the essential oils and acts as an antiplaque adjuvant [145]. Usage of ethanol in mouthwashes is very controversial and currently there are not many individual mouthwash products containing this solvent, but they are very popular and time-honoured. Ethanol has been replaced by propylene glycol in most brands, but the ethanol-containing mouthwashes remain in production and used in high volumes.

Ethanol itself is not genotoxic, mutagenic or carcinogenic. Nevertheless, acetaldehyde (ACH), the first metabolite of EtOH, is a carcinogen [146,147] and it has been observed that ACH levels may play an important role in the carcinogenesis of alcohol in the oral cavity [148]. Ingested alcohol is metabolised in the liver via ACH to acetate (by alcohol and aldehyde dehydrogenases), then to carbon dioxide and water; ACH in this process is degraded very efficiently, resulting in no increased ACH concentration in the blood [147]. Aldehyde dehydrogenase is essential to ACH elimination [149]. ACH from ethanol can be produced locally in the gastrointestinal tract by its epithelial cells as well as its flora (bacterial and fungal) [146,148,150,151]. Acetaldehyde can also be produced directly from glucose through the pyruvate-bypass pathway (by pyruvate decarboxylase) [149].

For an individual microbiome and its ability to produce ACH, poor oral hygiene, dysbiotic microflora (independent risk factor for oropharyngeal cancer), smoking and drinking alcohol increase the production of ACH [146,147,148,152]. Severe leukoplakia and insufficient oral hygiene promote a dysbiotic, fungal microflora [146,153]. Mutagenic ACH concentration can be produced by *Candida* yeasts either from ethanol (via alcohol dehydrogenases) or glucose (via the pyruvate-bypass pathway) [146]. Other potent producers of ACH are *Rothia mucilaginosa*, *Neisseria* spp., *Streptococcus* spp. [152,154]. A role of bacteria in carcinogenesis relies on the expression of alcohol dehydrogenases and the lack of ability to produce functional acetaldehyde dehydrogenases. If the bacteria have the gene encoding acetaldehyde dehydrogenase, they may have no carcinogenic effect or even possess a protective role [152]. Generally, the oral microbiome has a very low or no aldehyde dehydrogenase activity [147,149]. Therefore, mutagenic ACH concentration may be observed locally in saliva; in lower ethanol concentrations (ranging from 0.5‰ to 2.5%), mutagenic ACH concentration is observed as well [147].

While the correlation between the use of EtOH-based mouthwashes (in the absence of other factors) and the occurrence of head and neck carcinomas is unclear due to contradicting results across many studies [148], the use of ethanol and tobacco (both in smoke and smokeless form) together have the synergistic effect of significantly increasing the probability of carcinoma; other risk factors may potentiate the effect [148,155,156]. The time of exposure to carcinogens is vital to induce carcinogenesis. Nevertheless, ethanol-based mouthwashes usually contain additional antiseptics such as chlorhexidine (CHX), which is proposed to reduce prolonged mutagenic concentration of ACH when other carcinogenic risk factors are absent [147,148].

### 4.6. Propylene Glycol

Propylene glycol (propanediol, IUPAC propane-1,2-diol) is ubiquitous in topical and systemic drugs as well as in many foods, one of the most prevalent ingredients of mouthwashes. Due to a controversy of using ethyl alcohol in mouthwashes, or its side effects such as mucosa irritation, it was replaced mostly by propylene glycol. It is a viscous, faintly sweet, colourless alcohol containing two hydroxyl functional groups. It has been used as an emollient and solvent, more efficient than glycerine: it dissolves such compounds as phenols, vitamins A and D and many local anaesthetics. It also serves as a preservative. It is comparable to ethanol as an antiseptic [157]. Propylene glycol is generally recognised as safe. To our knowledge, there is no study investigating the impact of regular use on the oral tissues and properties of saliva. Propylene glycol is also an ingredient in the liquid used in electronic cigarettes, with no toxicologically relevant effects observed [144].

### 4.7. PEGylated Oils

PEGylated oils are derivatives of polyethylene glycol (PEG)—products of the etherification and esterification of glycerides and fatty acids with ethylene oxide. PEGylated oils function as a surfactant (emulsifying or solubilizing agents) used in cosmetics. They may also be used as non-ionic surfactants in oral, topical and parenteral drug delivery systems. There are a large number of oils and multiple PEG chain lengths [158].

PEG-40 hydrogenated castor oil (trade name Cremophor RH 40) is often used in mouthwashes. It is a polyethylene glycol derivative of hydrogenated castor oil where an average PEG chain is 40. Castor oil triglycerides are primarily composed of ricinoleic acid residues, approximately 13% are other acids; therefore, pegylated castor oil is a rather complex mixture of structurally related molecules [158]. PEG-40 hydrogenated castor oil is the most used PEGylated oil in mouthwashes as well as in cosmetics. PEG-40 hydrogenated castor oil functions as an emulsion stabiliser and surfactant. It helps solubilise fragrances and fat-soluble vitamins and may enhance hydrophilic penetration [158]. It shows a moderate cytotoxic effect (a negative impact on keratinocyte viability), but no antibacterial properties [159].

### 4.8. Zinc Compounds

Zinc is an essential trace element and its deficiency results in various pathologies or potentiates them [160,161]. Zinc ions are necessary for the activity of over three hundred enzymes in the human body [162]. Zinc compounds have a variety of positive effects on oral health but are not common in mouthwashes. As shown in our study, the following compounds of zinc are used in mouthwashes: zinc acetate, zinc chloride, zinc citrate, zinc gluconate, zinc hydroxyapatite, zinc lactate, zinc pidolate and zinc sulphate.

Zinc ions present antibacterial properties and are used as an antiplaque agent. The antibacterial property is a result of targeting the cytoplasm and glycolytic enzymes and the inhibition of glycolysis [38]. Zinc ions promote remineralisation and prevent dentin demineralisation, and also reduce enamel solubility via incorporation into hydroxyapatite surfaces [38]. Volatile sulphur compounds (VSC), such as hydrogen sulphide, methyl mercaptan or dimethyl sulphide, are responsible for malodour. Zinc ions, by antibacterial effect on VSC-producing bacteria as well as by direct inhibition on some VSC (by converting hydrogen sulphide and methyl mercaptan into non-volatile zinc sulphides), may help in reducing halitosis [163,164,165]. They also reduce calculus formation, by crystal growth modification or inhibition [166].

Matrix metalloproteinases (MMPs), responsible for the degradation of extracellular matrix proteins (e.g., collagen), play a significant role in periodontitis or dentine erosion [67,167]. MMPs are zinc- and calcium-dependent enzymes [168]. Potentially, oral topical substitution or binding of zinc (and calcium) ions may result in decreased activity of MMPs. Inhibition of MMPs by stannous ions and benefits of stannous fluoride usage were observed [67,127], but a zinc-binding system may pose a risk. As was said above, zinc ions provide many benefits. In addition, zinc ions are required for the activity of an enzyme crucial for oral acid-base homeostasis (carbonic anhydrase VI) [169]. Moreover, systemic zinc deficiency may result in a high caries prevalence [170,171]. Zinc deficiency may also be involved in the pathogenesis of common oral mucosal diseases (such as erosive oral lichen planus, burning mouth syndrome, recurrent aphthous stomatitis and xerostomia), and zinc supplementation may be useful in the prevention and treatment [172].

### 4.9. Limitations of the Study

There are several limitations of our study. There is no certainty that every single mouthwash product on the market has been found. There is also no certainty that an indicated property of components is observed in some or all mouthwashes, as for each composition there are many factors at play that may alter the characteristics of the substance (such as pH, solvents, etc., as mentioned in Section 3.1). Another limitation may be the composition list provided by the producer on the package (or website); we have not performed chemical analysis and rely solely on the accuracy of the above.

## 5. Conclusions

Despite there being numerous mouthwashes and countless components, there is not enough information about many of them. We recommend not to multiply individual mouthwash products and their ingredients beyond medical (or pharmaceutical) necessity, especially without scientific proof. 

Nowadays, there may be a great chance to provide novelties. Mouthwashes in the form of a tablet to prepare a fresh solution each time would help reduce plastic waste and take advantage of substances unstable in water solution, even when they are stored for months. Several well-known substances (such as sodium bicarbonate, stannous fluoride or filmogenic substances) have the potential for novel applications—those should be explored and the compounds possibly introduced in new mouthwash products. However, it is also vital to consider and study the long-term side effects of those substances.

## Figures and Tables

**Table 1 ijerph-19-03926-t001:** Ingredients of mouthwashes and their main functions therein.

Compound (In Alphabetical Order)	Main Functions or Commentary
1,3-propanediol	solvent
2-bromo-2-nitropropane-1,3-diol	preservative
2-propanol (isopropyl alcohol)	solvent, preservative
acesulfame K (potassium acesulfame)	sweetener
acidulated phosphate fluoride	anticaries mixture, consists of phosphoric acid, monosodium or disodium phosphate and sodium fluoride [15]
allantoin	unknown, uncertain oral topical meaning (wound healing)
aluminium lactate	astringent (causing shrinkage of the tissues, arrest of secretion, control of bleeding)
ammonium acryloyldimethyltaurate/VP copolymer	thickener, emulsion stabilising, film former (2)
amyloglucosidase (amylase)	an enzyme catalysing the hydrolysis of starch into sugars
anethole (^)	fragrance
arginine	anticaries protection, remineralisation (3) [16,17]
ascorbic acid (vitamin C vitamer)	antioxidant, uncertain oral topical meaning (antiplaque, tartar control, anti-inflammatory) [18]
aspartame	sweetener
benzoic acid	preservative
benzyl alcohol	preservative, fragrance
betaine (trimethylglycine, glycine betaine)	uncertain (humectant)
BHT	see ‘butylated hydroxytoluene’
bifida ferment lysate	uncertain (probiotic)
bisabolol (^)	anti-inflammatory, antimicrobial (4), fragrance
bromelain	anti-stain, dental plaque control [19]
buteth-3	solvent
butylated hydroxytoluene (BHT)	antioxidant
butylene glycol	solvent, humectant
caffeine	fragrance
calcium fluoride	anticaries protection, remineralisation (3)
calcium gluconate	anticaries protection, remineralisation (3) [20]
calcium glycerophosphate	anticaries protection, remineralisation (3) [21,22]
calcium lactate	anticaries protection, remineralisation (3) [20,23]
cannabidiol	anti-inflammatory, analgesic, antioxidant [24,25]
caprylyl glycol	humectant
caprylyl/capryl wheat bran/straw glycosides	surfactant (1)
caramel	fragrance, pigment
carbomer (polyacrylic acid)	thickener, emulsion stabilising, film former (2)
carvone	fragrance
cellulose gum (carboxymethyl cellulose)	thickener, emulsion stabilising, film former (2)
cetylpyridinium chloride	antimicrobial (4)
charcoal powder	pigment, uncertain oral topical meaning (bleaching property not observed, anti-odour) [26]
chitosan	film former (2), antiplaque [27]
chlorhexidine diacetate	antimicrobial (4)
chlorhexidine digluconate	antimicrobial (4)
chlorine dioxide	antimicrobial (4)
chlorobutanol	preservative
chlorphenesin	preservative
cinnamal (cinnamaldehyde) (^)	fragrance
citral (^)	fragrance
citric acid	buffering, pH adjuster (acidifier), chelating, component of effervescent salts when in the form of a tablet or powder (see ‘effervescent salts’)
citronellol	fragrance
CI (colour index), eg. Cl 42090	pigment
cocamidopropyl betaine	surfactant (1), thickener
colostrum	antibacterial, consists of immune components such as lactoferrin, lysozyme, lactoperoxidase, complement [28]
copper gluconate	uncertain (wound healing)
Cremophor RH 40	see ‘PEG-40 hydrogenated castor oil’
cyclodextrin	solubiliser, uncertain oral topical meaning (anti-odour) [29,30]
dehydroacetic acid	preservative
diazolidinyl urea	preservative (formaldehyde releaser)
dichlorobenzyl alcohol	antimicrobial (4)
diethylhexyl sodium sulfosuccinate	surfactant (1)
dipotassium glycyrrhizate	sweetener
dipotassium oxalate	dentin hypersensitivity treatment (oxalate ions by plugging dentinal tubules; potassium ions by nerve desensitisation)
disodium C12-14 pareth-2 sulfosuccinate	surfactant (1)
disodium EDTA	chelating, preservative synergist [31]
disodium phosphate	buffering, pH adjuster (alkaliser), component of acidulated phosphate fluoride (see ‘acidulated phosphate fluoride’)
disodium pyrophosphate	tartar control, buffering, chelating
effervescent salts	a powder mixture of sodium bicarbonate and citric acid or/and tartaric acid (in general), produce effervescence when mixed with water; provide quick production of solution, mask the unpleasant taste of many drugs [32,33]
epigallocatechin gallate	antioxidant
essential oil of cloves	see ‘*Eugenia caryophyllus* (*caryophyllata*) (clove/leaf/bud) oil’
ethanol (ethyl alcohol)	solvent, antimicrobial (4)
ethyl lauroyl arginate HCl	preservative
ethyl menthane carboxamide	cooling agent
ethylhexylglycerin	preservative
ethylparaben	preservative
eucalyptol (^)	antimicrobial (4), fragrance
*Eucalyptus globulus* (leaf) oil	essential oil of blue gum, consists of i.a. eucalyptol (1,8-cineole) (see ‘eucalyptol’) [34,35]
*Eugenia caryophyllus (caryophyllata)* (clove/leaf/bud) oil (essential oil of cloves)	essential oil of clove, consists of i.a. eugenol (see ‘eugenol’) [36,37]
eugenol (^)	antimicrobial (4), fragrance
flavouring agent (aroma)	fragrance (composition is usually a trade secret)
fluorinol	see ‘nicotinyl alcohol HF’
fluorohydroxyapatite	anticaries protection, remineralisation (3)
fructose	sweetener, humectant
fusel wheat bran/straw glycosides	surfactant (1), anti-foaming, solvent
gellan gum	thickener, emulsion stabilising, film former (2)
geraniol (^)	fragrance
gluconic acid	chelating
glucose oxidase	see ‘modified lactoperoxidase system’, catalyser of the oxidation of glucose to hydrogen peroxide and D-glucono-δ-lactone
glucose pentaacetate	see ‘modified lactoperoxidase system’, substrate (presumably) of glucose oxidase (see ‘glucose oxidase’)
glycerine (glycerol) (^)	solvent, thickener, humectant, sweetener
glyceryl caprylate	surfactant (1), preservative
*Glycyrrhiza glabra* (*Glycirrhizia glabra*) root extract	fragrance, anti-inflammatory, antimicrobial, ingredient of modified lactoperoxidase system (see ‘modified lactoperoxidase system’) [38]
hyaluronic acid	wound healing (postoperative, gingivitis, periodontitis, ulcers), humectant, film former (2) [39]
hydrated silica	thickener, emulsion stabilising
hydrochloric acid	pH adjuster (acidifier)
hydrogen peroxide	antimicrobial (4), teeth-whitening agent
hydrogenated castor oil (unidentified specimen)	surfactant (1)
hydrogenated starch hydrolysate	sweetener, humectant, thickener
hydrolysed hyaluronic acid	wound healing, humectant, film former (2)
hydroxyapatite	anticaries protection, remineralisation (3)
hydroxyethyl cellulose	thickener, emulsion stabilising, film former (2)
hypothiocyanite	antimicrobial (4)
imidazolidinyl urea	preservative (formaldehyde releaser)
inulin	sweetener (presumably), prebiotic
isomalt	sweetener, humectant
Kolliphor RH 40	see ‘PEG-40 hydrogenated castor oil’
lactic acid	preservative, pH adjuster (acidifier), fragrance enhancer, 5% mouthwash in treatment of recurrent aphthous ulcerations [40,41]
*Lactobacillus*	a genus of bacteria, uncertain oral topical meaning (probiotic)
lactoferrin	antimicrobial (4), ingredient of modified lactoperoxidase system (see ‘modified lactoperoxidase system’)
lactoperoxidase	see ‘lactoperoxidase system’ or ‘modified lactoperoxidase system’, catalyser of the oxidation of several substrates by hydrogen peroxide
lactoperoxidase system	antimicrobial (4), consists of lactoperoxidase, thiocyanate, hydrogen peroxide and oxidised product (hypothiocyanite [see ‘hypothiocyanite’]) [42,43]
lactose	sweetener, humectant
levulinic acid	uncertain (preservative)
limonene (^)	fragrance
linalool (^)	fragrance
lysozyme	antimicrobial (4)
macrogolglycerol hydroxystearate	see ‘PEG-40 hydrogenated castor oil’
magnesium aspartate	unknown
magnesium stearate	anticaking (mouthwashes in the form of powder)
magnesium sulphate	uncertain (viscosity controlling)
maltodextrin	sweetener
*Mentha arvensis* (leaf) oil	essential oil of wild mint, consists of i.a. menthol (see ‘menthol’) [44]
*Mentha piperita (*leaf*)* oil	essential oil of peppermint, consists of i.a. menthol (see ‘menthol’) [45,46]
*Mentha piperita* leaf extract	extract of peppermint, consists of flavonoids, i.a. catechin; naringenin (antioxidants) (added to distinct from *Mentha piperita* oil) [47]
*Mentha piperita* water	fragrance (added to distinct from *Mentha piperita* oil) [48]
*Mentha spicata* (herb) oil	essential oil of spearmint, consists of i.a. carvone (see ‘carvone’) and limonene (see ‘limonene’) (added to distinct from *Mentha piperita* oil and *Mentha arvensis* oil) [45,49]
*Mentha viridis* (leaf) oil	*Mentha spicata* subspecies (see *‘Mentha spicata*’) (added to distinguish from *Mentha piperita* oil) [50]
menthol (^)	antimicrobial (4), fragrance, cooling agent
menthol carboxamide	fragrance, cooling agent
menthone glycerine acetal	fragrance, cooling agent
menthyl lactate	fragrance, cooling agent
menthyl nicotinate	humectant, unknown oral topical meaning, ester of niacin and menthol
methyl salicylate (^)	anti-inflammatory, fragrance
methylisothiazolinone	preservative
methylparaben	preservative
Mg-Sr-carbonate hydroxyapatite conjugated with chitosan	anticaries protection, remineralisation (3); chitosan: film former (2) [51,52]
modified lactoperoxidase system	antimicrobial (4), consists of potassium thiocyanate, lactoperoxidase, glucose oxidase, glucose pentaacetate, lactoferrin, *Glycirrhizia glabra* root extract (check also ‘lactoperoxidase system’) [53]
neohesperidin di(hydro)chalcone	sweetener, fragrance enhancer
niacinamide (vitamin B3 vitamer)	unknown
nicomethanol hydrofluoride	see ‘nicotinyl alcohol HF’
nicotinyl alcohol HF (fluorinol, nicomethanol hydrofluoride)	anticaries protection, remineralisation (3) [54]
o-cymen-5-ol	preservative
octenidine dihydrochloride (octenidine HCl)	antimicrobial (4)
olaflur	anticaries protection, remineralisation (3)
panthenol (vitamin B5 vitamer)	humectant, uncertain oral topical meaning (moisturiser, wound healing)
papain	anti-stain, dental plaque control [19]
PEG-12 (polyethylene glycol)	humectant, solvent
PEG-12 allyl ether	unknown
PEG-12 dimethicone (polyether-group silicone polymer)	surfactant (1), emollient
PEG-32 (polyethylene glycol)	humectant, solvent
PEG-35 castor oil	surfactant (1)
PEG-40 hydrogenated castor oil (macrogolglycerol hydroxystearate, Cremophor RH 40, Kolliphor RH 40) (^)	surfactant (1)
PEG-60 hydrogenated castor oil	surfactant (1)
PEG-8 (polyethylene glycol)	humectant, solvent
pegylated hydrogenated castor oils (^)	surfactant (1)
pentasodium triphosphate	tartar control, buffering, chelating
pentylene glycol	solvent, humectant
phenethyl alcohol	preservative, fragrance
phenoxyethanol	preservative
phosphoric acid	pH adjuster (acidifier), buffering, component of acidulated phosphate fluoride (see ‘acidulated phosphate fluoride’)
phthalimidoperoxycaproic acid	teeth-whitening agent
poloxamer 188	surfactant (1)
poloxamer 338	surfactant (1)
poloxamer 407	surfactant (1), anti-foaming, thickener
polycarbophil	film former (2)
polyglycerin-10	humectant [55]
polyglyceryl-10 laurate	surfactant (1)
polyglyceryl-10 myristate	surfactant (1) [56]
polyglyceryl-10 stearate	surfactant (1)
polyglyceryl-3 caprate/caprylate/succinate	surfactant (1)
polyglyceryl-4 caprate	surfactant (1)
polyglyceryl-4 laurate/sebacate	surfactant (1)
polyglyceryl-5 oleate	surfactant (1)
polyglyceryl-6 caprylate/caprate	surfactant (1)
polylysine	preservative (if ε-polylysine)
polysorbate 20 (Tween 20)	surfactant (1)
polysorbate 80	surfactant (1)
polyvinyl methyl ether/maleic acid copolymer (PVM/MA copolymer)	film former (2), emulsion stabilising [57]
polyvinylpyrrolidone (PVP)	surfactant (1), thickener, emulsion stabilising, film former (2)
polyvinylpyrrolidone/vinyl acetate copolymer (PVP/VA copolymer, VP/VA copolymer)	film former (2), emulsion stabilising [58]
potassium chloride	dentin hypersensitivity treatment (nerve desensitisation)
potassium citrate (tripotassium citrate)	dentin hypersensitivity treatment (nerve desensitisation), buffering, chelating, pH adjuster (alkaliser)
potassium fluoride (KF)	anticaries protection, remineralisation (3), dentin hypersensitivity treatment (nerve desensitisation)
potassium nitrate	dentin hypersensitivity treatment (nerve desensitisation)
potassium phosphate (monopotassium phosphate)	buffering, uncertain oral topical meaning (nerve desensitisation)
potassium sorbate	preservative
potassium thiocyanate	see ‘thiocyanate’
propolis extract	wound healing, antioxidant, antimicrobial (4) [59,60]
propylene glycol (^)	solvent, antimicrobial (4)
propylparaben	preservative
PVM/MA copolymer	see ‘polyvinyl methyl ether/maleic acid copolymer’
PVP/VA copolymer	see ‘polyvinylpyrrolidone/vinyl acetate copolymer’
retinyl palmitate (vitamin A vitamer)	antioxidant, uncertain oral topical meaning (wound healing)
*Rosmarinus officinalis* (leaf) oil	essential oil of rosemary, consists of i.a. camphor, eucalyptol (1,8-cineole) (see ‘eucalyptol’) [61]
saccharin	see ‘sodium saccharin’
*Salvia lavandulaefolia* (leaf) oil	essential oil of Spanish sage, consists of i.a. camphor, borneol, α-thujone, eucalyptol (1,8-cineole) (see ‘eucalyptol’) [62,63]
*Salvia officinalis* oil	essential oil of common sage, consists of i.a. camphor, α-thujone, eucalyptol (1,8-cineole) (see ‘eucalyptol’) [63,64,65]
sodium acrylates/methacryloylethyl phosphate copolymer	film former (2)
sodium anisate	preservative
sodium ascorbyl phosphate	antioxidant, a stable precursor of vitamin C
sodium benzoate	preservative
sodium benzotriazolyl butylphenol sulfonate	UV light stabiliser
sodium bicarbonate (^)	buffering, pH adjuster (alkaliser, mild), component of effervescent salts when in the form of a tablet or powder (see ‘effervescent salts’)
sodium C14-16 olefin sulfonate	surfactant (1)
sodium C14-17 alkyl sec sulfonate	surfactant (1)
sodium carbonate	pH adjuster (alkaliser), buffering
sodium caseinate	unknown
sodium chloride	uncertain (taste, viscosity controlling)
sodium citrate	see ‘trisodium citrate’
sodium coco-sulphate	surfactant (1)
sodium cocoyl glutamate	surfactant (1)
sodium dehydroacetate	preservative
sodium ethylhexylglycerin	see ‘ethylhexylglycerin’
sodium ethylparaben	see ‘ethylparaben’
sodium fluoride	anticaries protection, remineralisation (3)
sodium gluconate	chelating
sodium hexametaphosphate	tartar control, chelating, emulsifier
sodium hyaluronate	wound healing, humectant, film former (2) [66]
sodium hydroxide	pH adjuster (alkaliser)
sodium hydroxymethylglycinate	preservative
sodium lactate	buffering, humectant
sodium lauroyl sarcosinate	surfactant (1)
sodium lauryl sulphate (sodium dodecyl sulphate)	surfactant (1)
sodium levulinate	uncertain (preservative)
sodium metabisulfite	preservative, antioxidant
sodium methyl cocoyl taurate	surfactant (1)
sodium methylparaben	see ‘methylparaben’
sodium monofluorophosphate (SMFP)	anticaries protection, remineralisation (3)
sodium myristoyl sarcosinate	surfactant (1)
sodium perborate	preservative, bleaching
sodium phosphate (monosodium phosphate)	buffering, component of acidulated phosphate fluoride (see ‘acidulated phosphate fluoride’)
sodium phytate	chelating, preservative synergist
sodium propionate	preservative
sodium propylparaben	see ‘propylparaben’
sodium saccharin	sweetener
sodium sulfate	uncertain (viscosity controlling)
sorbitol	sweetener, humectant, thickener
stannous chloride	antimicrobial (4), anti-erosive, inhibitor of metalloproteinases [38,67]
stannous fluoride (^)	anticaries protection, remineralisation (3), antiplaque, antimicrobial (4), tartar control, anti-odour, anti-erosive, inhibitor of metalloproteinases
steareth-30	surfactant (1) [68]
stevia	sweetener
*Stevia rebaudiana* leaf extract	see ‘stevia’
stevioside	the main sweetener found in the stevia leaves (see ‘stevia’)
strontium acetate	dentin hypersensitivity treatment (by plugging dentinal tubules)
sucralose	sweetener
tannase	anti-stain
tartaric acid	component of effervescent salts when in the form of a tablet or powder (see ‘effervescent salts’)
tetrapotassium pyrophosphate	tartar control, buffering, chelating
tetrasodium EDTA	chelating, preservative synergist [31]
tetrasodium glutamate diacetate	chelating
tetrasodium pyrophosphate	tartar control, buffering, chelating
thiocyanate	see ‘lactoperoxidase system’ or ‘modified lactoperoxidase system’, substrate of lactoperoxidase (see ‘lactoperoxidase’)
thymol (^)	antimicrobial (4), fragrance
*Thymus serpillum* (leaf) oil	essential oil of wild thyme, consists of i.a. thymol (see ‘thymol’) [69]
*Thymus vulgaris* (flower/leaf) oil	essential oil of common thyme, consists of i.a. thymol (see ‘thymol’) [70,71]
tocopheryl acetate (vitamin E vitamer)	see ‘vitamin E’
tributyl citrate	solvent
triclosan	antimicrobial (4)
trisodium citrate (sodium citrate)	buffering, chelating, pH adjuster (alkaliser)
ubiquinone (coenzyme Q)	antioxidant
vitamin E	antioxidant, uncertain oral topical meaning (wound healing)
VP/VA copolymer	see ‘polyvinylpyrrolidone/vinyl acetate copolymer’
water	solvent
xanthan gum	thickener, emulsion stabilising, film former (2)
xylitol	sweetener, anticaries protection, remineralisation (3), humectant [72]
zinc acetate (^)	anti-odour, dental plaque control, tartar control
zinc chloride (^)	anti-odour, dental plaque control, tartar control
zinc citrate (^)	anti-odour, dental plaque control, tartar control
zinc gluconate (^)	anti-odour, dental plaque control, tartar control
zinc hydroxyapatite (^)	anticaries protection, remineralisation (3), anti-odour, dental plaque control, tartar control [73,74]
zinc lactate (^)	anti-odour, dental plaque control, tartar control
zinc PCA (zinc pidolate) (^)	anti-odour, dental plaque control, tartar control, humectant
zinc sulphate (^)	anti-odour, dental plaque control, tartar control

(1) Surfactant (may act as a solubiliser, emulsifier, emulsion stabiliser, detergent, wetting agent, foaming agent/defoamer and may possess antimicrobial features) [75]; (2) mucoadhesive film former (filmogenic agents) (it is suggested that they may act as a moisturiser, as dressing for wounds, to prevent dental plaque and stain formation, to prevent demineralisation, as a barrier to relieve pain in oral lesions such as aphthous ulcers, to improving the delivery and retention of actives) [27,76,77,78,79,80,81,82,83]; (3) remineralisation agent (to remineralise the demineralised tooth surface in dental caries or erosion as well as to prevent demineralisation, also in dentin hypersensitivity to obturate the dentinal tubules); (4) antimicrobial agent (an oral antiseptic, it may act as a preservative); (^) means it is discussed in the subsequent text.

**Table 2 ijerph-19-03926-t002:** Analysis of selected mouthwash components.

Group	Among All Mouthwashes (1)		Within a Group (2)					
	*n*	%	Component	*n*	%	Component	*n*	%
**identified ** **specimens and their form**	241	100.0%	aqueous solutions or concentrates	234	97.1%	tablets etc. for making aqueous solutions	7	2.9%
**fluorine compounds**	153	63.5%	olaflur	18	11.8%	nicotinyl alcohol HF	3	2.0%
			fluorohydroxyapatite	2	1.3%	sodium monofluorophosphate	10	6.5%
			sodium fluoride	133	86.9%	potassium fluoride	2	1.3%
			stannous fluoride	2	1.3%	calcium fluoride	1	0.7%
**concentration of fluoride ions (ppm)**	153	63.5%	98–135	15	9.8%	187–250	102	66.7%
			400–500	18	11.8%	900–1136	8	5.2%
			1450	1	0.7%	N/A	9	5.9%
**arginine**	3	1.2%						
**sodium bicarbonate (3)**	3	1.2%						
**potassium compounds (4)**	43	17.8%	dipotassium oxalate	2	4.7%	potassium citrate	3	7.0%
			potassium fluoride	2	4.7%	potassium chloride	10	23.3%
			potassium phosphate	5	11.6%	potassium nitrate	9	20.9%
			tetrapotassium pyrophosphate	20	46.5%			
**zinc compounds**	47	19.5%	zinc acetate	5	10.6%	zinc chloride	13	27.7%
			zinc citrate	10	21.3%	zinc gluconate	8	17.0%
			zinc hydroxyapatite	2	4.3%	zinc lactate	6	12.8%
			zinc PCA	4	8.5%	zinc sulphate	1	2.1%
**aluminium lactate**	4	1.7%						
**phosphorus and calcium compounds (5)**	43	17.8%	calcium gluconate	1	2.3%	calcium glycerophosphate	4	9.3%
			calcium lactate	2	4.7%	fluorohydroxyapatite	2	4.7%
			hydroxyapatite	8	18.6%	zinc hydroxyapatite	2	4.7%
			Mg-Sr-carbonate hydroxyapatite conjugated with chitosan	2	4.7%	tetrapotassium pyrophosphate	20	46.5%
			disodium pyrophosphate	2	4.7%	tetrasodium pyrophosphate	16	37.2%
			pentasodium triphosphate	7	16.3%	sodium hexametaphosphate	4	9.3%
**stannous compounds**	4	1.7%	stannous chloride	2	50.0%	stannous fluoride	2	50.0%
**antimicrobial drugs (6)**	126	52.3%	chlorhexidine	50	39.7%	essential oils (7)	113	89.7%
			cetylpyridinium chloride	84	66.7%	octenidine HCl	1	0.8%
			dichlorobenzyl alcohol	1	0.8%	hydrogen peroxide	2	1.6%
			chlorine dioxide	5	4.0%	triclosan	1	0.8%
**essential oils**	114	47.3%	thymol (8)	30	26.3%	menthol (9)	90	78.9%
			eucalyptol (10)	28	24.6%	methyl salicylate	18	15.8%
			eugenol (11)	21	18.4%	bisabolol	7	6.1%
**chlorhexidine compounds**	50	20.7%	digluconate	49	98.0%	diacetate	1	2.0%
**concentration of chlorhexidine (%)**	50	20.7%	0.05	6	12.0%	0.06	3	6.0%
			0.09	1	2.0%	0.1	5	10.0%
			0.12	7	14.0%	0.2	13	26.0%
			N/A	15	30.0%			
**ethanol**	26	10.8%						
**glycerine**	180	74.7%						
**propylene glycol**	103	42.7%						
**surfactants**	223	92.5%	PEG-40 hydrogenated castor oil	114	51.1%	poloxamer 407	55	24.7%
			sodium lauryl sulphate	25	11.2%	polyvinylpyrrolidone	11	4.9%
			polysorbate 20	45	20.2%	cocamidopropyl betaine	13	5.8%
			others (12)	78	35.0%			
**sweeteners**	233	96.7%	sorbitol	89	38.2%	xylitol	87	37.3%
			stevia (13)	11	4.7%	lactose	1	0.4%
			maltodextrin	7	3.0%	saccharin	144	61.8%
			neohesperidin dichalcone	3	1.3%	aspartame	1	0.4%
			fructose	1	0.4%	hydrogenated starch hydrolysate	13	5.6%
			acesulfame K	22	9.4%	sucralose	39	16.7%
			isomalt	7	3.0%	dipotassium glycyrrhizate	2	0.9%
			glycerine	180	77.3%			
**preservatives (14)**	198	82.2%	2-bromo-2-nitropropane-1,3-diol	13	6.6%	benzoic acid	17	8.6%
			benzyl alcohol	22	11.1%	ethylhexylglycerine	7	3.5%
			lactic acid	14	7.1%	methylparaben	37	18.7%
			phenoxyethanol	11	5.6%	potassium sorbate	57	28.8%
			propylparaben	23	11.6%	sodium benzoate	127	64.1%
			others (12)	31	15.7%			
**colourants**	159	66.0%						
**flavouring and cooling agents (15)**	239	99.2%						

(1) Among all mouthwashes (total number or percentage of mouthwashes containing indicated groups of ingredients among all mouthwashes); (2) within a group (expanded when a group consists of more than one element; total number or percentage of mouthwashes containing indicated ingredients within a particular group); (3) sodium bicarbonate (it is not counted if other components of effervescent salts are present and if the product is a mouthwash in the form of a tablet or powder to dissolve in water); (4) potassium compounds (except potassium sorbate, potassium thiocyanate, dipotassium glycyrrhizate, acesulfame K); (5) phosphorus and calcium compounds (except sodium ascorbylphosphate, sodium monofluorophosphate, sodium acrylates/methacryloylethyl phosphate copolymer); (6) antimicrobial drugs (except peptides and proteins, preservatives, ethanol and other alcohols, sodium bicarbonate, stannous compounds, zinc compounds, fluorine compounds); (7) essential oils (thymol, menthol, eucalyptol, eugenol, bisabolol or oils with an expected high concentration of them, not methyl salicylate); (8) thymol and/or *Thymus serpillum* (leaf) oil and/or *Thymus vulgaris* (flower/leaf) oil (counted once for one individual product, if more than one present); (9) menthol and/or *Mentha arvensis* (leaf) oil and/or *Mentha piperita*(leaf) oil (counted once for one individual product, if more than one present); (10) eucalyptol and/or *Eucalyptus globulus* (leaf) oil and/or *Rosmarinus officinalis* (leaf) oil and/or *Salvia lavandulaefolia* (leaf) oil and/or *Salvia officinalis* oil (counted once for one individual product, if more than one present); (11) eugenol and/or *Eugenia caryophyllus* (*caryophyllata*) (clove/leaf/bud) oil (essential oil of cloves) (counted once for one individual product, if more than one present); (12) others (according to Table 1 and Appendix A); (13) stevia and/or stevioside and/or *Stevia rebaudiana*leaf extract; (14) preservatives (antimicrobial agents except ethanol and other alcohols, essential oils, chlorhexidine, cetylpyridinium chloride, propolis extract, lactoferrin, dichlorobenzyl alcohol, hydrogen peroxide, lactoperoxidase system, lysozyme, octenidine dihydrochloride, chlorine dioxide, triclosan); (15) flavouring and cooling agents (counted with essential oils); N/A—not available.

## Data Availability

Data are contained within the article and Appendix A.

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
