# Peer review of "A Fresh Look at Mouthwashes—What Is Inside and What Is It For?"

_ijerph, 2022, doi:10.3390/ijerph19073926_

Round 1
Reviewer 1 Report
First of all, I would like to thank the Editor for the possibility of reviewing this article entitled “A fresh look at mouthwashes – What´s inside and what´s it for? This paper is in the field of International Journal of Environmental Research and Public Health. It addresses an interesting topic; however, I would recommend the authors several modifications before considering its publication.
Title:
It is original, but I suggested to add that this article investigates polish mouthwashes market.
*The “citation” at the left side of the first page is incomplete.
Abstract:
I suggest to modify “or” for “and” in line 15.
Conclusion (lines 21-22) should be modified. It does not provide enough information about real conclusions of the study.
Introduction:
Line 40: Delete “)” after “Brushing”.
Line 47: The sentence cannot be finished with an “etc”. I recommend to modified it as follows: “antiplaque substances prevent as well as support treatment of periodontal diseases, among others.”
Lines 51-52: In which circumstances mouthwashes are used in Esthetic Dentistry? I suggest modifying this sentence.
Material and Methods:
Line 87: Authors have specified that they searched for mouthwashes available on the market in Poland, but how was the searched conducted?
Lines 91-92: Why you did not contact the producer to request missing information about mouthwashes´ composition?
Line 98: In formal English you should not use verbal contractions (e.g., We´ve).
Results and Discussion
Table 1: I suggest to add it as a supplementary file.
Avoid use question marks when you do not know the property of a compound (e.g., allantoin: wound healing?
Can table 1 be summarized? E.g., calcium fluoride, calcium gluconate, calcium glycerophosphate, calcium glycerophosphate and calcium lactate have anticaries protection and remineralisation properties. All these compounds can be summarized as “calcium derivates”. A way to summarize this table is matching up compounds by main functions. Check it.
“3.1. Components” and “3.2. Analysis of mouthwash components” are the same. In addition, lines 130 and 131 should be at the beginning of the “results and discussion” section.
Results and discussion section may be divided. Discussion should start from 3.3. section. In this regard, from lines 195 to 480 should be summarized. A discussion about other similar articles should be added.
Conclusions:
Why mouthwashes may be avoided when oral hygiene is insufficient? In oral surgeries, chlorhexidine is usually used as a chemical cleaning in the immediate postoperative instead of mechanical cleaning. I suggest deleting lines 482 to 483.
The last part of the conclusions section can be added as “Recommendations for further investigations” at the end of the discussion section.
References
The article is well documented with a vast number of references.
Reviewer 2 Report
Major comments
I also think that the topic may be of interest to the field.
Although this manuscript demonstrates a large amount of data, I have the following concerns.
â‘ The authors should compare the results with those of the previous studies.
â‘¡Overall, the content is verbose, so please correct it to a clear description. Organizing is necessary about the table likewise.
â‘¢Please clearly state the novelty of this study.
â‘£The description of the "method" of the abstract is not enough.
Reviewer 3 Report
The article reports on ingredients of mouthwashes sold over the counter in Poland.
In the abstract, the aim stated was to investigate the compositions of mouthwashes, their functions, effectiveness in preventing and curing oral diseases, or side effects; while in the introduction the aim was to investigate the compositions of mouthwashes and their functions. After reading through the paper, I only find that the paper was to investigate the composition of the mouthwashes. Perhaps the author should change their study aims while the function, effectiveness and side effects are what the authors discuss.
The abstract also mentioned about statistical analysis which was not mentioned in the methods.
"We searched for mouthwashes available on the market": please specify the the place.
"We've analysed only products available on the market as of the time of conducting this study": please state when the study was undertaken.
Table 2: please separate footnote from text. It was confusing to read.
In results and discussion, please discuss the limitations of the study.
Conclusions need to be revised and relate to the aim of the study. Perhaps suggest the clinical significance of the outcome of your findings.
Supplement: There are many names, are these the trade names? Perhaps a column of the manufacturer would be relevant as trade names can change.
Round 2
Reviewer 1 Report
Dear authors,
Congratulations for the modifications. In spite of that, i recommend that "3.3. Limitations of the study should be moved to the end of the "discussion" section.
Best wishes.
Author Response
Dear Reviewer,
thank you for your kind review again and acceptance.
The limitations has been moved, according to the suggestion.
Sincerely
Reviewer 2 Report
All the points raised in the first version of the manuscript have been properly addressed by the authors.
Author Response
Dear Reviewer, thank you for your kind review and acceptance.
This manuscript is a resubmission of an earlier submission. The following is a list of the peer review reports and author responses from that submission.